# Influence of Nutrition on Mental Health: Scoping Review

**DOI:** 10.3390/healthcare11152183

**Published:** 2023-08-01

**Authors:** Lara María Suárez-López, Lluna Maria Bru-Luna, Manuel Martí-Vilar

**Affiliations:** 1Department of Basic Psychology, Faculty of Psychology and Speech Therapy, Universitat de València, 46010 Valencia, Spain; larasuarez@hotmail.com; 2Department of Education, Faculty of Social Sciences, Universidad Europea de Valencia, 46010 Valencia, Spain

**Keywords:** mental health, nutrition, feeding, anxiety, depression, psychopathology

## Abstract

The aim of this article was to carry out a scoping review of existing research on the influence of food on mental health: (1) Background: nutrition, nutrient levels or an adequate body weight seem to influence the mental health status of individuals. The consumption of psychotropic drugs also seems to contribute to overweight; (2) Methods: fifteen previous research articles were used in the review, which were read in their entirety, following PRISMA methodology and using SPIDER and GRADE tools; (3) Results: there is a relationship between diet and mental health status. Low levels of magnesium, together with high levels of calcium, provoke anxious states, and supplementation with Melissa Officinalis attenuates them. Healthy nutritional habits generally reduce depressive symptoms; while vitamin D supplementation improves mental health status; (4) Conclusion: in general, healthy and appropriate nutrition, such as the Mediterranean diet, improves mental health status. Levels of magnesium, vitamin D, and vitamin B6 also seem to have an influence. As limitations of the present review, “food” was considered any form of nutrient administered as an independent variable, and it may include linguistic and publication bias.

## 1. Introduction

The foods that are available and consumed today, which can be found in almost any supermarket in the world, are more varied than ever and come from a multitude of places. The purpose of this work is to provide more scientific knowledge about our eating habits and foods, considering their relationship or consequences on mental health. It also provides a possible starting point for further scientific research. Such knowledge has repercussions on a better relationship between people and the way they eat, acquiring greater awareness when choosing what to eat and knowing how it will influence the organism and mental health. Several contributions on the relationship between mental health and nutrition are included.

### 1.1. Theoretical Framework

#### 1.1.1. Mental Health

The World Health Organization [1] defines mental health as:

A state of mental well-being that enables people to cope with stressful times in life, to develop their full abilities, to be able to learn and work adequately, and to contribute to the betterment of their community. Mental health is also a fundamental human right. And an essential element for personal, community and socioeconomic development (p. 1).

When the diet lacks the necessary nutrients that contribute to the normal, non-pathological functioning of the organism, this can lead to the development of inappropriate mental health, especially when talking about anxiety or mood disorders, such as depression and/or increased levels of stress. In this sense, inflammation is a normal reaction of the body’s immune system that serves to combat actual or potential physical discomforts (toxic molecules; [2]). Analogous to physical health, inflammatory processes also appear to be related to mental health. For example, chronic low-grade inflammation causes deleterious effects on overall mental health status [3].

The response mechanism in anxiety disorders begins when the symptomatology provoked by that anxiety becomes “pathological” or “maladaptive” and starts to become a disproportionate response or one that occurs in the face of a non-existent threat [4,5]. On the other hand, the behavioral and emotional consequences of depressive processes also have a utilitarian origin [6]. However, that would cease to be functional when it interferes in the areas of the person’s life, either because of its intensity, duration, frequency, etc.

#### 1.1.2. Nutrition

On the other hand, it is believed that changes in nutrition may be related to changes in the microbiota (or gut) and, therefore, in the brain.

The human brain requires carbohydrates, proteins, fats, vitamins and minerals to function properly. It is the most demanding organ because, although its approximate weight is equivalent to only 2% of the total body weight, it nevertheless requires 20–30% of the energy consumed in a day [7]. Its main fuel is glucose, which comes mostly from carbohydrate intake, and to a lesser extent from a process called non-glucose precursors or “gluconeogenesis” (which literally means “generation of new glucose” [7]). It is known that low carbohydrate intake does not appear to have an impact on emotions in the short term, although it does appear to do so in the long term. This could be due to reduced serotonin levels [7]. For example, suffering from diabetes (i.e., exceeding the optimal glucose level) shows a correlation with a higher incidence of depressive disorders. In contrast, levels below the optimal glucose level (as in reactive hypoglycemia) produce typical symptoms of anxiety disorders (choking, palpitations, feeling dizzy or fainting, numbness in the extremities, etc. [8]).

Proteins are made up of basic units called amino acids, whose main function is the neuronal transfer of information, i.e., they are the substances by which information is transported from one neuron to another [7]. Some of these amino acids are precursors of the most important neurotransmitters related to higher processes or psychological functions. Table 1 explains how three amino acids influence psychological processes in a simple way.

Within fats, only omega-3 and omega-6 (alpha-linolenic acids) are considered essential fatty acids. Both have a fundamental role (anti-inflammatory and pro-inflammatory characteristics) in the activated brain response to any threat, also called “neuroinflammation.” The omega-3 DHA (docosahexaenoic acid), in addition, would be related to neuronal protection and neurotransmission because it contributes to the maintenance of the functional integrity of the neuron, improves the sensitivity of neuroreceptors and the fluidity of phospholipid membranes [7]. On the other hand, within fatty acids, there are several types: saturated and transmonosaturated fatty acids (proinflammatory), and polyunsaturated fatty acids (anti-inflammatory). Foods such as whole grains, fish or fruits and vegetables, characterized by containing polyunsaturated fatty acids, are associated with a protective function of the organism [9].

Likewise, the body also requires vitamins and minerals for its proper functioning. They are considered “essential nutrients” and are involved in various nerve functions and muscle development. They must be consumed through fruits and vegetables because human beings do not have the capacity to generate them [10].

The shortage or deficit of all these nutrients, as well as certain gastrointestinal alterations that affect the body’s microbiota, may contribute to the triggering of an inflammatory defense mechanism of the gut itself. This would generate greater sensitivity to stress and would occur similarly in processes of anxiety or depression, derived from the bidirectional relationship of the gut-brain axis [11,12].

The intestinal barrier plays a crucial role in preventing an excessive inflammatory response contrary to the microbiota itself. If there is a failure in the regulation of this inflammatory response, it could cause inflammatory phenomena detrimental to intestinal and/or systemic health [3]. Likewise, it should be noted that dietary supplements have also been shown to be effective on cognitive function and mental health status. For example, vitamin supplements are able to reverse depressive symptoms, as seen in patients with Alzheimer’s disease. In addition, various types of vitamins and minerals are associated with a restructuring of mood and memory processes [13].

#### 1.1.3. Psychopharmacology

Regarding psychotropic drugs (antidepressants, anxiolytics, benzodiazepines, antipsychotics, etc.), there may be a lack of control or weight gain (overweight or obesity) even in those who, at the beginning of treatment, are normal weight. This could worsen the general state of their health or lead to a lack of adherence to the treatment itself, resulting in relapse for these people [14]. This fact was investigated in a study with mental health patients in a center aimed at the psychosocial field in Brazil. The study detailed that overweight is higher in patients who used serotonin reuptake inhibitor antidepressants, typical and atypical antipsychotics, and benzodiazepines [15].

Despite this, it has been found in the field of Psychiatry that the combination of psychopharmacology, together with psychotherapy, physical activity, and nutritional interventions, is important to contribute to the biopsychosocial model of mental illness [16]. This model considers the state of health as the consequence of the interaction between biological (or organic), psychological and social facts [17].

#### 1.1.4. Physical Exercise

Physical exercise causes energy consumption or energy expenditure through movement of the body. Such physical activity, whether at a medium or intense level, improves overall health. It can also contribute to a better quality of life, mental health or greater well-being [18].

### 1.2. Justification

The need to carry out this review was due to the concern that arises to know the degree of the relationship between mental health and nutrition. Nutrition and lifestyle have been found to be determined by the following factors [19]: people’s status and purchasing power [20], irregularity in routines (different schedules among family members and habits such as “snacking” or eating between meals [21]), precooked foods and fast food (increasingly frequent in Western society, which leads to a scarce variety of nutrients [21]) and little exercise (the levels of physical activity are increasingly lower, favoring a greater sedentary lifestyle [22]).

In order to know the state-of-the-art of this topic, we conducted an initial search in Cochrane, a database composed mainly of systematic reviews in the field of health sciences and found very few relevant synthesis studies. Of the 20 that were found using the keywords “nutrition mental health”, only two of the systematic reviews were focused on nutrition and mental or psychosocial health. The first [23] was published more than 10 years ago and focused only on a very specific age range and population. The second [24] was published more recently but focused on the same population and age range. This shows the importance and the need for further research on this topic due to a clear knowledge gap. Furthermore, authors such as Andreo-Martínez et al. [25] state that this is “one of the most interesting topics in biomedical research during the last few years”.

Finally, this work also relates to the Sustainable Development Goals of the United Nations Organization, specifically Goal 3: Good Health and Well-being, which aims to ensure healthy lives and promote well-being for all at all ages.

### 1.3. Objectives

The general objective of the present study is:To analyze the connection between nutrition and mental health in individuals through a scoping review of the scientific literature.

The specific objectives are:To analyze the relationship between nutrition and anxiety, depression, stress and insomnia.To analyze the influence of different foods, food supplements/supplements or nutrients on the organism and their subsequent impact on mental health.

## 2. Materials and Methods

To develop this work, an exhaustive literature search was carried out in relation to the connection between food or nutrition and mental health. This was followed by a careful and critical reading of the studies on this subject. The whole process was carried out following the PRISMA methodology to ensure the systematicity of the study [26] (Appendix A). In addition, since this study focuses on the search for evidence from qualitative or mixed research, the SPIDER tool was used [27], based on the same principles as the PICO tool. It was considered necessary to introduce a recommendation on clinical practice, so the GRADE methodology was used to optimize the evaluation of the quality of the evidence and the grading of the strength of the recommendations [16].

### 2.1. Eligibility Criteria

Prior to the selection of articles, the following eligibility criteria were defined. The search protocol, including these criteria, was registered in openscienceframework (OSF) with the identification code of osf.io/95n2z.

#### 2.1.1. Inclusion Criteria

The following criteria were considered for the inclusion of articles:(a)Scientific articles of empirical research or intervention.(b)Articles that included samples of people from the general or specific population.(c)Articles investigating the relationship between nutrition and mental health.(d)Articles of qualitative, quantitative, or mixed type.(e)Articles published between 2008 and 2022 (inclusive).(f)Articles published in Spanish or English.(g)Articles that include protocols for subsequent randomized trials.

#### 2.1.2. Exclusion Criteria

As exclusion criteria, in addition to those derived from the previous inclusion criteria, are discarded:(a)Articles published in journals belonging to pseudoscientific areas (e.g., parapsychology, paramedicine, etc.).(b)Articles belonging to congresses or conferences.(c)Single case studies.(d)Articles derived from the legal field.

### 2.2. Information Sources

As mentioned above, first, an initial general search was carried out in Cochrane. Next, the inclusion and exclusion criteria were detailed, and finally, a more specific search was carried out, selecting those articles considered valid. The documentary search of the individual studies was carried out in January 2023 in two of the main scientific search engines: PubMed and Mendeley.

### 2.3. Search

Searches were carried out in all the search engines mentioned above, always relating a series of terms in Spanish and/or English using the Boolean operator AND. Table 2 shows the combination of terms used for each database.

### 2.4. Study Selection Process

Once the search for articles had been carried out, 390 articles were obtained, of which 134 were found in PubMed and 256 in Mendeley. A first screening was performed by reading the title and abstract of the articles. The articles that met the eligibility criteria went on to a second phase in which the complete body of the article was read. They were downloaded into the Mendeley Bibliography Manager, and an Excel document was created to organize and analyze the documents found. Those articles that did not meet the eligibility criteria were excluded from the review.

This process was carried out by one of the authors and corroborated by another through the Covidence tool.

### 2.5. Coding

The variables included in the review and their coding were as follows. As independent variables:The consumption or deficit of some nutrients, foods, supplements or food supplements.The practice of some type of healthy diet.Health care, professional advice, or nutritional education.

The dependent variables included in the review were:Decrease or aggravation of mental health problems or disorders.Decrease or worsening of other pathologies other than mental health.

### 2.6. Data Extraction Process

The data were extracted using an Excel table, in which the following factors were included:Authors and year of publication.Phenomenon of interest.Type of research.Research design.Sample.Results obtained.Limitations.

## 3. Results

### 3.1. Selection of Results

Figure 1 shows the entire article selection process.

The identification phase included those terms included in the keywords, as well as in the title and abstract, in PubMed and Mendeley. A total of 390 potential studies were found and screened for inclusion in the review: the title and abstract were read, and those that were considered not to meet the study criteria were eliminated (*n =* 334). Of those 334 studies found in both search engines, 288 were duplicates.

In the eligibility phase, the remaining papers (*n =* 56) were evaluated for eligibility. After reading the completed studies, 41 were eliminated for not showing a relationship between mental health and nutrition (*n =* 17), for having been published in languages other than Spanish and English (*n =* 3), for being single-case studies (*n =* 12), for deriving from the legal field (*n =* 1), for belonging to congresses or conferences (*n =* 5) or for belonging to pseudoscientific research (*n =* 3).

In the last phase, the studies that would form part of the present work were determined. Finally, a total of 15 articles were included in the scoping review.

### 3.2. Characteristics of the Included Studies

Table 3 below presents each study included in the review, as well as its most important characteristics.

### 3.3. Summary of the Studies

As a synthesis of the studies found, in relation to the phenomenon of interest, *n =* 8 of the fifteen studies included in the present scoping review are focused on depressive disorder (in any of its degrees), *n =* 2 focus on depressive disorder together with anxious or stressful disorders, *n =* 1 includes samples of people with polycystic ovary syndrome (PCOS), *n =* 1 of people with obesity, *n =* 1 in situation of pregnancy and obesity treated with probiotics, *n =* 1 with ADHD and *n =* 1 with magnesemia.

Regarding the type of research, *n =* 1 is a qualitative study or recommendation for clinical practice, *n =* 1 is a prospective follow-up study, and *n =* 13 are randomized clinical trials.

Specifically, in the first article [28], a clinical practice guideline is presented that included specific dietary recommendations or nutritional guidelines for the treatment or prevention of depressive disorder. The guideline includes five dietary recommendations: consumption of traditional “Mediterranean”, “Norwegian”, or “Japanese” diet, increased intake of fruits and vegetables, inclusion of foods rich in omega-3 in the diet, replacement of unhealthy foods with more nutritious options, and limitation of consumption of processed foods. In addition, it was found that Nutritional Psychiatry maintains that specific dietary patterns may influence the risk of suffering from depression. On the other hand, Pinto-Sanchez [29] studied the influence of a probiotic NCC3001 on anxiety, depression, and irritable bowel syndrome (IBS). The results concluded that such a probiotic had a significant positive effect on depression and an increase in quality of life. However, it had no effect on anxiety or IBS. Changes in brain activity were also found, as the probiotic reduced limbic activity.

In the third study, by Von Berens et al. [30], quality of life was related to health in older adults and the absence of depressive symptoms. All participants performed physical exercise for 2 to 3 weeks and were divided into experimental (with probiotic treatment) and placebo groups. Regardless of belonging to either of these two groups, the mental component improved in all of them, as well as depressive symptoms. However, no changes in the physical component were recorded. Haybar et al. [31], for their part, described that patient with cardiovascular disease suffered from depressive symptomatology, anxiety, stress, or sleep disturbances. The aim of the study was to relate the consumption of Melissa Officinalis (MO), known for its calming effect, with the reduction in the symptomatology described above. The results were positive for the experimental group compared to the placebo group.

In the fifth study, Forsyth et al. [32] reported the importance of diet and exercise in patients undergoing treatment for depression, anxiety, or stress in Primary Care (PC). They were divided into two groups: one received dietitian sessions and exercise or sport physiology sessions, the other received scheduled telephone contact. Both groups obtained improvement for anxious, depressive or stress symptomatology and in the Australian Modified Healthy Eating Index (Aus-HEI) total scores. However, improvements in quality of life were more likely to be sustained in the group that worked with dietitians, thus highlighting the importance of working together with dietitians for the treatment of anxiety, depression, or stress.

Jacka et al. [33], in the sixth study, investigated the efficacy of dietary changes or nutritional guidelines on the treatment of existing mental illnesses, specifically in the treatment of major depression. The results showed that dietary improvement may provide an effective and accessible treatment strategy for the management of this highly prevalent mental disorder, the benefits of which could be extended to the management of common comorbidities. On the other hand, Ostadmohammadi et al. [34] related the effect of co-administration of vitamin D and probiotics on mental health in women with POS. They obtained beneficial effects overall on mental health parameters, total serum testosterone, hirsutism, high sensitivity for C-reactive protein, total plasma antioxidant capacity, total glutathione, and malondialdehyde levels.

Kazemi et al. [35] built on the relationships found in other research between alterations in the gut microbiota and the pathophysiology of depression, as well as the effect that probiotics and, in some cases, prebiotics can have on the gut microbiota. They compared two groups whose participants showed depressive symptomatology (analyzed using the Beck Depression Inventory): one of them received treatment with probiotics, and the other with prebiotics. Improvement was seen only in the group that received probiotics. In the ninth study, Parletta et al. [36], investigated the effect of a Mediterranean diet with fish oil supplementation on depressive symptomatology in adults. They concluded that changes in nutritional patterns are achievable, and that fish oil supplementation could alleviate the negative effects of depressive disorder.

Meanwhile, Uemura et al. [37] worked on the basis that gut microbiota is related to obesity and mental health status. They concluded that nutrition education on gut microbiota in Japanese women with obesity could improve obesity and psychological factors. As for the study by Dawe et al. [38], the authors started from the assumption that poor mental health in pregnancy causes problems at this stage and in the child, in addition to obesity worsening mental health and physical health in expectant mothers. The aim of their article was to study whether probiotics improved mental health in pregnant women with obesity. They concluded that in all groups, there was an increase in anxiety and physical worsening, and in addition, no beneficial results that contributed to mental well-being were found.

On the other hand, Kaviani et al. [39] studied the effects of vitamin D supplementation on the severity of depressive symptoms, based on studies linking low vitamin D levels with a dysregulated hypothalamic-pituitary-adrenal (HPA) axis and depressive symptomatology. The results showed that such supplementation was beneficial in reducing depressive symptomatology. The work of Hemamy et al. [40] analyzed the effect of vitamin D and magnesium supplementation on mental health in children with ADHD. The experimental group received vitamin D and magnesium supplementation, while the placebo group did not receive supplementation.

The results showed that the combination with supplementation could have beneficial effects on the improvement of behavioral function in children with ADHD, as well as on their mental health.

Noah et al. [41], on the other hand, reported that the level of magnesium and vitamin B6 is related to mental health and/or quality of life. They worked with two groups: one group received combined supplementation of vitamin B6 and magnesium, and the other group received only magnesium. Both groups obtained positive results for the reduction in anxiety, depression, or stress, although the improvements were greater in the combined supplementation group for daily physical activity. Finally, Roponen et al. [42] assumed that there are studies that inversely relate diet quality to the severity of depressive disorder. In their work, they argued the influence of nutritional education and social support intervention for the treatment of depressive symptomatology.

In addition, the scoping review provides an overview of the topics covered, as well as the common similarities and differences between the articles. 

Common similarities:

Several articles focus on the impact of dietary supplementation on mental health and well-being, including the effects of vitamin D, probiotics, magnesium, and omega-3.

Randomized controlled trial (RCT) designs are utilized in some studies to evaluate the effects of specific interventions on populations with mental disorders such as depression and anxiety.

A shared concern for the mental well-being of specific populations is evident, including pregnant women, older adults, individuals with metabolic disorders (such as obesity and polycystic ovary syndrome), and patients with chronic conditions.

Some studies also explore the effects of dietary and lifestyle interventions on quality of life and mood.

Highlighted differences:

Each article focuses on a specific aspect of mental health and employs different interventions, such as probiotics, nutritional supplements, dietary changes, or a combination thereof.

The studies address various mental disorders, including depression, anxiety, mood disorders, and gastrointestinal disorders.

Certain articles specifically target unique populations, such as women with polycystic ovary syndrome, individuals with metabolic disorders, or children with attention-deficit hyperactivity disorder (ADHD).

The study designs may vary, encompassing diverse sample sizes, intervention durations, and outcome measures.

It is important to note that this comparison is general and based on an overview rather than a detailed analysis of the content of each article. For more precise and comprehensive information regarding specific similarities and differences, it is recommended to review each article individually.

## 4. Discussion

The general objective of this scoping review is to analyze the existing scientific literature that studies the relationship between nutrition and mental health. It is also considered important to take the influence of nutrition on the intestinal microbiota into account and, as a consequence, on the brain and mental health.

To begin with the interpretation of the results, it is possible to state that there is a relationship between the state or quality of mental health and the intake of nutrients and/or supplementation, whether based on some type of vitamin or mineral, or probiotics [28,29,31,32,33,34,35,36,39,40,41,42]. This may be due to the relationship between reduced levels of magnesium, together with elevated levels of calcium and stress, which generate agitation, anxiety, hallucinations, confusion, asthenia, insomnia, delirium and/or headache, among others [43].

On the other hand, anxiety has been reduced in patients with chronic stable angina who consumed supplementation [31] and in patients who received nutritional education [32]. However, it has not been reduced with the probiotic BL in patients with IBS [29]. Such improvement could be perceived since awareness or submission to an individualized self-management system is related to improved diet, compliance with medical visits and pharmacological follow-up, as well as general improvements in functional status [44].

Favorable research was found for the relationship between healthy changes in nutritional habits and a decrease in depressive symptoms [28,29,31,32,33,35,36,39,42]. There was a single study in which there was no significant improvement in depressive symptomatology in the sample due to supplementation [30]. Notably, a significantly positive result was found in a sample of adults with IBS [29]. In general, this could be due to the fact that balanced nutrition is necessary for good brain function. Also, deficient levels of omega-3 fatty acids are related to an increased risk of depression, in addition to deficits in other parameters such as iron, folate or vitamin B12, among others [45].

Regarding the relationship of probiotics with the improvement of the organism and mental health, a slight favorable result for the decrease in depressive disorders is seen in two studies [29,34], contrary to another study [38]. In addition, such improvement was found in a particular mental health group in women with PCOS [34]. This fact could be related to decreased oxidative stress and improved insulin function [46].

As for treatment with prebiotics, no relationship is found between them and improvement in overall mental health status [35]. This underscores the need for further research with prebiotics in humans. Animal studies yielded favorable results for rats with prebiotic treatment and anxiolytic effects on gut microbiota proliferation [47].

Regarding studies dealing with the influence of vitamin D supplementation on mental health, an improvement in mental health was observed in all investigations [34,39,40]. Furthermore, this was reflected in a specific improvement of behavioral function and mental health status in child population with ADHD [40]. The scientific literature suggests that this may be since compensated levels of vitamin D in the brain are needed for proper functioning, and that deficient levels are related to some form of depression [48].

Similarly, including magnesium supplementation seems to improve stress levels when combined with vitamin B6 [41]. Likewise, a study was found in which consumption of the painkiller MO improved depressive or anxious symptomatology, as well as stress levels or sleep disorders [31]. This could be since the effects on the organism are comparable to those obtained by pharmaceutical anxiolytics [49].

Another study in which nutrition or healthy eating habits were theoretically accounted for as an independent variable in those investigations also found or predicted that this would contribute to positive outcomes in mental health status [28,37,42]. One of them was directly a clinical practice recommendation guideline [28]. This may be because the more “western” the diet is, the higher the risk for depression, contrary to what happens when it is mostly Mediterranean [50].

The studies included in the review have several limitations, among which the following stand out. For instance, the need to conduct research with larger samples than those found, the specificity of these samples, and the duration of some studies or their cost-effectiveness. Finally, the fact that they were conducted from a mainly psychiatric field, requiring further research from the psychological field.

As limitations of the present scoping review, firstly, the term “food” was used to refer to any form of nutrient administered as an independent variable. Most of the studies worked with the incorporation of food supplementation or complementation (vitamin D, vitamin B6, magnesium, etc.). In addition, it would be interesting for future research to use more databases to continue research on the study, and articles in languages other than Spanish and English could be included to reduce linguistic bias. Finally, only articles published in peer-reviewed journals were included, producing the so-called publication bias.

As implications of this study, it is important to keep in mind that, although there is evidence of the benefits of probiotics for mental health, there is still much that is unknown about exactly how they work and which strains and doses are the most effective. In addition, it is worth considering that probiotics are not a substitute for conventional treatments for mental health and should not be used as a replacement for proper medical care. Future research is needed with real foods, not supplementation/complementation, to study their effect on mental health, as the relationship between the microbiota-brain axis appears to have a high correlation with mental health balance. Further research on the improvement of digestive processes, immune function, and the consequent reduction in mental illnesses would also seem to be desirable.

In any case, it would be very difficult to point out nutrition as the only cause of mental health status. This status is influenced by various causes that are multifactorial themselves. The state of mental health cannot be separated from nutrition, but neither can psychological problems or disorders be blamed solely on nutrition. Other factors such as the economic level or inflation processes, the lifestyle (in this case mostly sedentary), the educational and cultural level of society, the integral crises that arise today or politics and the scarcity of resources, among others, cannot be disregarded.

## 5. Conclusions

It is concluded that, in general, food has an influence on the state of mental health in people. The consumption of probiotics improves psychological health, although more research is needed. It is necessary to know the exact strains and effective doses. In any case, they do not replace current pharmacological treatments for mental health disorders, they could be used in a complementary way. More research is also needed to know the impact of real foods on psychological state. The state of mental health cannot be separated from food, but neither can psychological problems or disorders be blamed solely on food.

## Figures and Tables

**Figure 1 healthcare-11-02183-f001:**
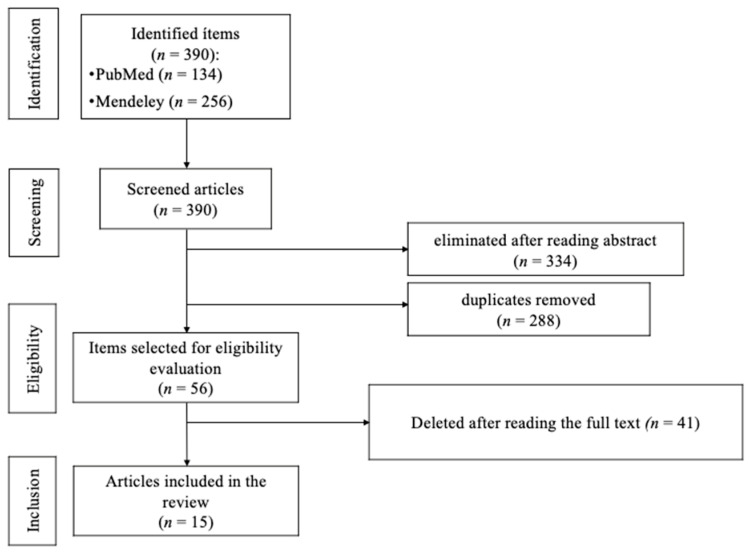
Flow chart according to PRISMA.

**Table 1 healthcare-11-02183-t001:** Influence of amino acids at the psychological level.

Amino Acid	Role in Neurotransmission	Psychological Function	Fortified Foods	Cognitive-Emotional Consequences
Tryptophan	Serotonin precursor	Emotional balance, sociability, lividity, and sleep (melatonin synthesis)	Foods of animal origin, soybeans and derivatives, bananas, avocados, nuts, etc.	Depression, anxiety, insomnia, and EDCs
Histidine	Histamine precursor	Sleep and sexual orgasm	Food of animal origin	Insomnia and anorgasmia
Tyrosine	Precursor of dopamine, adrenaline, and norepinephrine	Mental energy, impulse control, motivation, alertness, memory, executive functions, and processing speed	Food of animal origin, whole grains, and legumes	Addictions, apathy, weight gain, Parkinson’s, depression, and loss of performance

Source: adapted from Baldó [7].

**Table 2 healthcare-11-02183-t002:** Synthesis of search strategies.

Search Equation	Database	Results
“nutrition” AND “mental health” “depression” AND “nutrition” “anxiety” AND “nutrition” “microbiota” AND “salud mental”	PubMed	134
“depression” AND “nutrition” “anxiety” AND “nutrition” “salud mental” AND “microbiota” “trastornos psicológicos” AND “salud mental”	Mendeley	256

Source: own elaboration.

**Table 3 healthcare-11-02183-t003:** Characteristics of the included studies.

Authors and Year of Publication	Objective	Phenomenon of Interest	Type of Research and Design	Sample	Results	Limitations
Opie et al., 2016 [28]	To provide a set of practical measures based on the best available evidence to inform clinical and public health recommendations	Major depressive disorder	Qualitative recommendation for clinical practice Not reported	Not reported	To reduce depression, it is recommended to follow traditional dietary patterns such as the Mediterranean, Norwegian or Japanese diet, increase the consumption of fruits, vegetables and legumes. Also, including foods rich in polyunsaturated acids, and eliminate the consumption of unhealthy foods	Not reported
Pinto- Sánchez et al., 2017 [29]	To evaluate the effects of Bifidobacterium longum NCC3001 (BL) on anxiety and depression in patients with IBS	IBS and depression-anxiety	Prospective follow-up study Double-blind, experimental group BL or placebo group, in IBS and anxious-depressive symptomatology	44 adults with IBS and anxious-depressive symptomatology	14/22 patients in the BL group had a reduction in depression scores vs. 7/22 patients in the placebo group. BL had no significant effect on anxiety or IBS symptoms.	Not reported
Von Berens et al., 2018 [30]	To analyze the effects of a physical activity program in combination with protein supplementation on health-related quality of life (HRQoL) and depressive symptoms in community-dwelling, mobility-limited older adults	Depressive disorder/depressive symptomatology	Randomized clinical trial Double-blind, G1: nutritional supplementation + physical exercise G2: no nutritional supplementation + physical exercise	149 seniors	There was a significant improvement in both the mental component summary and the Center for Epidemiological Studies Depression Scale (CES-D) during the intervention. No differences were detected between those who received the nutritional supplement and those who received the placebo. The results did not reveal significant changes in the Physical Component Summary (PCS) or variations in the effects between the subcategories	Not reported
Haybar et al., 2018 [31]	To investigate the effect of Melissa officinalis (MO) capsules on depression, stress, anxiety, and sleep disorders of patients with chronic stable angina (CSA)	Depression, anxiety, stress and insomnia in patients with CSA	Randomized clinical trial Double-blind, G1: intervention with calming MO G2: placebo	80 patients with ASC	The intervention group receiving MO capsules had a significant reduction in scores of depression, anxiety, stress, and total sleep disturbance, compared with the placebo group. Eight-week supplementation with 3 g of MO can decrease these conditions in patients with CSA	Not reported
Forsyth et al., 2015 [32]	To evaluate the efficacy of a diet and exercise lifestyle intervention on mental health outcomes for patients currently being treated for depression and/or anxiety in primary care	Depressive or anxious disorder	Randomized clinical trial G1: individualized care in primary care G2: control group without such care	119 patients with depressive or anxious disorder	Significant improvement was found for both groups on Depression, Anxiety and Stress Scale (DASS) scores, measures of nutrient intake and total Australian modified Healthy Eating Index (Aust-HEI) scores. Significant differences between groups over time were found only for iron intake and body mass index. Patients participating in individual consultations with a dietitian were more likely to maintain or improve diet quality than those participating in an attention control	Not reported
Jacka et al., 2017 [33]	To investigate the efficacy of a dietary program for the treatment of major depressive episodes. In this trial, Supporting the Modification of lifestyle in Lowered Emotional States (SMILES)	Major depressive disorder	Randomized clinical trial Single-blind, parallel groups, G1: seven sessions of individualized nutritional counseling. G2: social support group of the same duration as G1	116 patients	The dietary support group demonstrated significantly greater improvement between baseline and 12 weeks on the Montgomery–Åsberg Depression Rating Scale (MADRS) than the social support control group Improved diet may provide an effective and accessible treatment strategy for the management of this highly prevalent mental disorder	Not reported
Ostadmohammadi et al., 2019 [34]	To determine the effect of vitamin D and probiotic co-administration on mental health, hormonal, inflammatory and oxidative stress parameters in women with polycystic ovary syndrome (PCOS)	PCOS and MH	Randomized clinical trial Double-blind experimental group vitamin D + probiotic and placebo group	60 subjects between 18 and 40 years of age	Vitamin D and probiotic co-supplementation, compared with the placebo, significantly improved beck depression inventory, general health questionnaire scores and depression, anxiety and stress scale scores. Compared with the placebo levels, vitamin D and probiotic co-supplementation was associated with a significant reduction in total testosterone, hirsutism, high-sensitivity C-reactive protein and malondialdehyde (MDA) and a significant increase in total antioxidant capacity (TAC) and total glutathione (GSH) levels	Not reported
Kazemi et al., 2019 [35]	To compare the effect of supplementation with the probiotic and prebiotic on the Beck Depression Inventory (BDI) score as a primary outcome as well as the kynurenine/tryptophan ratio and tryptophan/branch chain amino acids (BCAAs) ratio as secondary outcomes in patients with major depressive disorder (MDD)	Major depressive disorder	Randomized clinical trial Double-blind, G1: probiotic G2: prebiotic G3: placebo	110 patients with depressive disorder 81 completed the trial	Probiotic supplementation resulted in a significant decrease in BDI score compared to the placebo and prebiotic supplementation. Inter-group comparison indicated no significant differences among the groups in terms of serum kynurenine/tryptophan ratio and tryptophan/BCAAs ratio. However, the kynurenine/tryptophan ratio decreased significantly in the probiotic group compared to the placebo group after adjusting for serum isoleucine.	Not reported
Parletta et al., 2019 [36]	To investigate the impacts of a Mediterranean-style diet intervention for mental health and quality of life (QoL) in people with depression using a RCT design over 3 months with follow-up at 6 months	Depressive disorder	Randomized clinical trial G1: Mediterranean diet education + fish oil supplementation 6 months G2: bi-weekly social groups 3 months	152 adults with self-reported depression	At 3 months, the MedDiet group had a higher MedDiet score, consumed more vegetables, fruit, nuts, legumes, wholegrains, and vegetable diversity; less unhealthy snacks and red meat/chicken. The MedDiet group had greater reduction in depression and improved mental health QoL scores at 3 months. Improved diet and mental health were sustained at 6 months. Reduced depression was correlated with an increased MedDiet score, nuts, and vegetable diversity. Other mental health improvements had similar correlations, most notably for increased vegetable diversity and legumes	Not reported
Uemura et al., 2019 [37]	To investigate changes in obesity and psychological health in obese Japanese women following nutritional education focusing on gut microbiota composition	Obesity and MH	Randomized clinical trial G1: experimental group (*n* = 22) G2: control group (*n* = 22)	44 women with obesity	After the intervention, dietary fiber intake, frequency of vegetable consumption, and frequency of milk and milk product consumption increased significantly in the intervention group, compared with the control group. Body weight and body mass index, waist circumference, and the depression scale score decreased significantly, while significant improvements were found in self-rated health and microbiome diversity	Not reported
Dawe et al., 2020 [38]	To investigate the influence of Lactobacillus rhamnosus GG and Bifdobacterium lactis BB12 on depression, anxiety, and functional health and well-being, among a multi-ethnic sample of pregnant women with obesity residing in the Counties Manukau Health region in South Auckland, New Zealand	Obesity and consequences in MH	Randomized clinical trial Double-blind experimental group with probiotics and placebo control group	230 pregnant women with obesity in New Zealand	Depression scores remained stable and did not differ between the probiotic and placebo groups at 36 weeks. Anxiety and physical well-being scores worsened over time irrespective of group allocation, and mental well-being scores did not differ between the two groups at 36 weeks. Probiotics did not improve mental health outcomes in this multi-ethnic cohort of pregnant women with obesity	Need for further studies to validate the findings
Kaviani et al., 2020 [39]	To explore the effect of high-dose vitamin D supplementation on depression, neurotransmitters, and HPA axis	Mild-to-moderate depressive disorder	Randomized clinical trial Double-blind, vitamin D experimental group and placebo control group	56 subjects mild to moderate depression	Following intervention, significant changes were observed in the intervention group compared to the controls: 25(OH)D concentrations increased and BDI scores decreased. Oxytocin concentrations were significantly reduced in controls, but between-group differences were insignificant. Within- and between-group differences of platelet serotonin concentrations were not significant; however, the increment in controls was higher	The duration of the study may not reflect the long-term effects of vitamin D on depression
Hemamy et al., 2021 [40]	To determine the effect of vitamin D and magnesium supplementation on mental health in children with ADHD	Children with ADHD	Randomized clinical trial with baseline and endline questionnaire on mental health Double-blind, experimental group with magnesium + vitamin D and control group with placebo	66 children with ADHD	After eight weeks of intervention, the serum levels of 25-hydroxy-vitamin D3 and magnesium increased significantly in the intervention group compared with the control group. Also, children receiving vitamin D plus magnesium showed a significant reduction in emotional problems, conduct problems, peer problems, prosocial score, total difficulties, externalizing score, and internalizing score compared with children treated with the placebo	Need for more studies with larger sample size
Noah et al., 2021 [41]	To explore whether addition of vitamin B6 to magnesium supplementation enhances any observed effects on mental health and quality of life	Low magnesemia and severe/extremely severe stress	Randomized clinical trial Blinded investigator, parallel groups; G1: vitamin b6 + magnesium G2: magnesium alone	74% women 26% men	DASS anxiety and depression scores significantly improved from baseline to week 8 with both treatments, particularly during the first 4 weeks. Improvement in quality of life continued over 8 weeks. Participants’ perceived capacity for physical activity in daily life showed greater improvement with magnesium + vitamin B6 than magnesium alone	Not reported
Roponen et al., 2021 [42]	To investigate the effectiveness and cost-effectiveness of a behavioral nutrition group intervention compared to a social support intervention in the treatment of depression	Depressive disorder/depressive symptomatology	Randomized clinical trial G1: 72 patients FM nutritional intervention G2: 71 patients social support group intervention	144 patients with mild or moderate depression and outpatient treatment	Improvement is anticipated in both groups, as the intervention would prove to be cost-effective and acceptable, can be implemented in health care to support the treatment of depression	Regarding profitability

Note. ADHD: Attention Deficit Hyperactivity Disorder. Aus-HEI: Australian Modified Healthy Eating Index. BCAAs: tryptophan/branch chain amino acids. BDI: Beck Depression Inventory. BL: Bifidobacterium longum NCC3001. CES-D: Center for Epidemiological Studies Depression Scale. CSA: Chronic Stable Angina. DASS: Depression, Anxiety and Stress Scale. FM: Food for Mind. G1: group 1. G2: group 2. GHS: glutathione. HPA: hypothalamic-pituitary-adrenal. HRQoL: health-related quality of life. IBS: Irritable Bowel Syndrome. MADRS: Montgomery–Åsberg Depression Rating Scale. MDA: malondialdehyde. MDD: Major Depressive Disorder. MedDiet: Mediterranean diet. MH: Mental Health. MO: Melissa Officinalis. PCS: Physical Component Summary. PCOS: Polycystic Ovary Syndrome. QoS: quality of life. SMILES: Supporting the Modification of lifestyle In Lowered Emotional States. TAC: total antioxidant capacity.

## Data Availability

The raw data supporting the conclusions of this article will be available from the authors upon request.

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
