# Peer review of "Influence of Nutrition on Mental Health: Scoping Review"

_healthcare, 2023, doi:10.3390/healthcare11152183_

Round 1
Reviewer 1 Report
In this review, authors summarized the influences of nutrition on mental health. They found that a healthy and appropriate nutrition can improve mental health status. This is a thorough review, but some issues need to be addressed.
1. Line 68: In this part, authors mainly focused on proteins and fats? What about other nutrients?
2. Line 122: Why did authors mention SARS-CoV-2 pandemic here? Is it related with nutrition and mental health?
3. In “3.3. Summary of the studies”, authors introduced some of the studies. It is suggested that authors can compare the studies and analyze their differences in this part.
4.
4. The language of the manuscript needs to be improved.
Moderate editing of English language required
Author Response
In this review, authors summarized the influences of nutrition on mental health. They found that a healthy and appropriate nutrition can improve mental health status. This is a thorough review, but some issues need to be addressed.
- Line 68: In this part, authors mainly focused on proteins and fats? What about other nutrients?
We have considered and modified your recommendation. We have included other types of nutrients, as vitamins and minerals, which are also considered as important for the correct function of the organism.
- Line 122: Why did authors mention SARS-CoV-2 pandemic here? Is it related with nutrition and mental health?
We have probably not properly explained the relationship between nutrition and mental health, that is why we have decided to delete this paragraph due to its inadequacy.
- In “3.3. Summary of the studies”, authors introduced some of the studies. It is suggested that authors can compare the studies and analyze their differences in this part.
We appreciate your contribution. It has been taken into account in order to conduct a comparison and a general analysis of the included articles has been made.
- The language of the manuscript needs to be improved.
All your comments, included the ones regarding language, has been considered. We consider they have highly improved the quality of our work, thank you for that.

Reviewer 2 Report
This review article summarizes the relationship between mental health (from psychosomatic disorders to psychosis) and nutrition (including vitamins). Pre-probiotics (including dysbiosis) are also mentioned as relevant factors. The article's content is consistent with conventional knowledge and is manageable in terms of content. As an educational article, publication of this article in a journal is appropriate.
 However, the following corrections are necessary.
1) In L24-43 of 3.3. Summary of the studies (Refs. 30 and 32), there is a reference to physical exercise. However, whether exercise is, an independent or confounding factor for mental health is not mentioned. Therefore, adding references 30 and 32 and summarizing the article as a review is inappropriate.
2) Table 3 does not include the number of the literature.
3) The year of publication of Forsyth's paper in Table 3 differs from Reference 32.
4) The year of publication of Jacka's paper in Table 3 differs from Reference 33.
Author Response
This review article summarizes the relationship between mental health (from psychosomatic disorders to psychosis) and nutrition (including vitamins). Pre-probiotics (including dysbiosis) are also mentioned as relevant factors. The article's content is consistent with conventional knowledge and is manageable in terms of content. As an educational article, publication of this article in a journal is appropriate.
 However, the following corrections are necessary.
1) In L24-43 of 3.3. Summary of the studies (Refs. 30 and 32), there is a reference to physical exercise. However, whether exercise is, an independent or confounding factor for mental health is not mentioned. Therefore, adding references 30 and 32 and summarizing the article as a review is inappropriate.
A new paragraph at the introduction section has been included in the manuscript thanks to this comment. It justifies the relevance of physical exercise or physical activity. Thank you for your comment.
2) Table 3 does not include the number of the literature.
Thank you very much, the numbers of each reference have been added in the table next to the name of the author and the year of each study.
3) The year of publication of Forsyth's paper in Table 3 differs from Reference 32.
We have made a mistake while transferring the data. We kindly appreciate your feedback.
4) The year of publication of Jacka's paper in Table 3 differs from Reference 33.
This appreciation is similar to the one at comment 3. We highly appreciate it.

Reviewer 3 Report
The manuscript entitled: “Influence of Nutrition on Mental Health: Scoping Review” addresses a topic of interest in the health area, namely the relation of nutrition and mental health, reviewing the literature published. I think it can give a contribution to the area. Nevertheless, I would like to make some remarks that I consider important prior to be published:
-Authors should use more concise language, with shorter sentences, to be easier the reading.
-The abstract should clearly distinguish the results and conclusions and include a reference to the limitations of the study.
-The introduction should be shortened. Most of the information of the introduction is general considerations that are not necessary.
-The 1.2 section (justification) and 1.3 (objectives) should be more concise and objective.
- Authors should clarify the including/excluding criteria: Only peer-reviewed journals were selected? What groups of population were studied? What is the relevance of including scientific publications about protocols?
-The table of the results could be more informative. Thus, a reduction in the text of results section could be possible, making the reading of the manuscript easier.
-What was the rationale to include a paper on qualitative recommendations for clinical practice (paper 1)?
-Lines 152 to 155 of the discussion are duplicated
-The main information obtained from the literature refers to supplementation, so, the introduction/discussion of the manuscript, including Table 1, should be reconsidered.
-English language needs moderate editing.
Author Response
The manuscript entitled: “Influence of Nutrition on Mental Health: Scoping Review” addresses a topic of interest in the health area, namely the relation of nutrition and mental health, reviewing the literature published. I think it can give a contribution to the area. Nevertheless, I would like to make some remarks that I consider important prior to be published:
-Authors should use more concise language, with shorter sentences, to be easier the reading.
Thanks for your appreciation. The sentences have been shortened to facilitate the reading.
-The abstract should clearly distinguish the results and conclusions and include a reference to the limitations of the study.
Thank you, we consider that the conclusions and results are clearly separated, but it is true that a reference to the limitations of the study, present in the Discussion, was missing. We have added it following your recommendation.
-The introduction should be shortened. Most of the information of the introduction is general considerations that are not necessary.
Thank you very much for your consideration, we have shortened and synthesized the introduction.
-The 1.2 section (justification) and 1.3 (objectives) should be more concise and objective.
The justification and objectives have also been synthesized according to your commentary and are now more concise.
- Authors should clarify the including/excluding criteria: Only peer-reviewed journals were selected? What groups of population were studied? What is the relevance of including scientific publications about protocols?
Thank you for your comment. As it can be read in materials and methods, PubMed and Mendeley databases were reviewed, so only peer-reviewed journals were selected. In addition, we did not narrow by population groups. Finally, we did not prioritize the inclusion of scientific publications on protocols, since these were also included.
-The table of the results could be more informative. Thus, a reduction in the text of results section could be possible, making the reading of the manuscript easier.
Thank you for your recommendation. Indeed, we have included a column with the objective of the study, which helps considerably to contextualize it, and we have increased the information in the results column. On the other hand, the results in text have been reduced and some studies have been compared with others.
-What was the rationale to include a paper on qualitative recommendations for clinical practice (paper 1)?
We believe it makes sense as it is a recommendation for clinical practice regarding diet as a preventive or protector resource against depression.
-Lines 152 to 155 of the discussion are duplicated
We appreciate your feedback, the duplicate information has been deleted.
-The main information obtained from the literature refers to supplementation, so, the introduction/discussion of the manuscript, including Table 1, should be reconsidered.
We acknowledge your detailed comment, we have included a new paragraph about supplementation at the section Nutrition, which is located at the Introduction.
Comments on the Quality of English Language
-English language needs moderate editing.
English has been reviewed and improved, we hope it is more understandable now, thank you very much.

Reviewer 4 Report
In this review, the authors highlighted the importance of food on mental health. The authors have employed PRISMA, SPIDER and GRADE tools for the 15 articles. The author’s observation of mediterranean diet improving mental health status is interesting. Table-1 summarized the association of amino acids with psychological function. Overall, the review is interesting and will benefit the readers largely.
The authors need to address the following for final consideration of this review article.
1. The article needs an abbreviation section.
2. An overall schematic diagram will be useful for readers.
Author Response
The authors need to address the following for final consideration of this review article.
- The article needs an abbreviation section.
Thank you for your comment, we have sent a list of all abbreviations as supplementary material to the publisher.
- An overall schematic diagram will be useful for readers.
Thanks to your comment, a brief diagram has been included to direct the readers about what can be found in this manuscript, thank you very much.

Round 2
Reviewer 1 Report
In this review, authors summarized the influence of food on mental health.The authors have revised the manuscript accordingly. The quality of the article has been improved.
Minor editing of English language required